# Immunogenicity Parameters of Cancer Patients Receiving the mRNA Vaccine BNT162b2 While Obtaining Radiotherapy: A Longitudinal Cohort Evaluation

**DOI:** 10.3390/vaccines12030275

**Published:** 2024-03-06

**Authors:** Paul Thöne, Margot Egger, Marija Geroldinger-Simic, Harald Kindermann, Lukas Kocik, Nicola Karasek, Barbara Fischerlehner, Kurt Spiegl, Georg Gruber, Bernhard Aschacher, Benjamin Dieplinger, Martin Clodi, Hans Geinitz

**Affiliations:** 1Department of Radiation Oncology, Ordensklinikum Linz Barmherzige Schwestern, 4010 Linz, Austria; 2Medical Faculty, Johannes Kepler University, 4020 Linz, Austria; 3Department of Laboratory Medicine, Konventhospital Barmherzige Brüder Linz and Ordensklinikum Linz Barmherzige Schwestern, 4010 Linz, Austria; 4Department of Dermatology, Ordensklinikum Linz Elisabethinen, 4020 Linz, Austria; 5Department of Marketing and Electronic Business, University of Applied Sciences Upper Austria, 4400 Steyr, Austria; 6Department of Medicine, Konventhospital Barmherzige Brüder Linz, Seilerstätte 2, 4020 Linz, Austria; 7CICMR—Clinical Institute for Cardiovascular and Metabolic Research, Johannes Kepler University, 4020 Linz, Austria

**Keywords:** seroconversion, humoral immune response, kinetic, antibody persistence, side effects, COVID-19

## Abstract

Background: Cancer patients are highly prone to infectious diseases. While undergoing antineoplastic treatment, the risk of severe symptoms upon infection increases, necessitating efficient protective measures, such as vaccination. For patients receiving radiotherapy, there is no specific information about humoral immunity. During the COVID-19 pandemic, serial antibody measurements were therefore offered to cancer patients, following SARS-CoV-2 vaccination while obtaining radiotherapy. Methods: Out of 74 enrolled patients, 46 met the inclusion criteria. Two cohorts were allocated, depending on an association with chemotherapy or pure radiotherapy. An additional healthy control cohort of 16 healthcare workers was enrolled. All participants followed a two-fold BNT162b2 vaccine schedule. SARS-CoV-2 binding antibodies were measured serially in a 7-day cycle for 35 days and over the long-term, using the Elecsys^®^ Anti-SARS-CoV-2 immunoassay. Results: Cancer patients under pure radiotherapy have a comparable humoral vaccination response and long-term persistency of antibodies to healthy controls. Patients receiving additional chemotherapy show a significantly delayed immune response and decreased antibody titers. The vaccine was well tolerated in all cohorts. Conclusions: Pure radiotherapy in cancer patients does not interfere with the vaccine-induced humoral immune response or other immunogenetic aspects, whereas previous or simultaneous chemotherapy does. Findings are of particular relevance for future epidemic or pandemic scenarios.

## 1. Introduction

Cancer patients have an increased susceptibility to infectious diseases [1]. This issue became particularly apparent during the coronavirus disease 2019 (COVID-19) pandemic, caused by the beta coronavirus SARS-CoV-2, which often caused severe symptoms primarily affecting the respiratory tract [2]. Infected cancer patients are highly prone to adverse disease progress [3]. This risk is even more pronounced in patients currently undergoing cancer treatment [4]. Consequently, there is a high necessity to immunize cancer patients. SARS-CoV-2 mRNA vaccines, as a form of active immunization, have shown a high level of efficacy and safety in preventing clinically severe COVID-19 infections [5]. In most patients, vaccination efficacy correlates with humoral immune response (HIR), expressed as the level of binding antibody (bAB) titer [6], and bAB response could be used to measure the efficiency of vaccination [7]. A high level of vaccination-induced bAB is associated with higher protection [8]. Vaccination during the administration of antineoplastic therapies, especially chemotherapy (CTh), is challenging due to its multifactorial immunosuppressive side effects of the CTh [9]. Concerning vaccination success under radiotherapy (RT), only limited data are available. Nevertheless, cancer patients are at increased risk and require prophylactic measures against infection, especially during cancer treatment. Therefore, in the pandemic scenario of COVID-19, there was an indication for cancer patients to immunize against SARS-CoV-2.

However, the immunization of cancer patients and during radiotherapy is challenging for multiple reasons. Cancer creates an immunosuppressive cellular environment and inhibits immune cell division and lymphocyte proliferation [10]. Adverse parameters such as the diagnosis of metastatic disease, advanced age, male gender, and hematologic malignancies impair HIR of SARS-CoV-2 vaccination [11]. Moreover, there is an impairment of HIR in cancer, depending on the underlying oncological entity [12]. Essentially, cancer treatments influence proper immune responses. Conventional CTh causes substantial immunosuppressive side effects [13]. Specific targeted therapy, as well as CTh and steroids, interferes with the formation of bAB [11]. The impact of RT on the immune system, even in local treatment, has controversial effects, addressing various molecular signaling pathways, molecular patterns, and mediators [14]. All of these findings are important for combination perspectives with antineoplastic immunotherapies. RT has immediate side effects on the immune system, such as bone marrow suppression and lymphopenia up to an extended period [9]. Depending on the area of radiation, RT damages immune-related stem cells and releases immunosuppressive mediators, such as adenosine, VEGF A, chemokines and TGF-β [15]. Simultaneously, antibody formation is mediated by B cells and T cells [16]. The differentiation and division of T cells and dendritic cells can be suppressed through RT [17]. RT can moreover interfere with adaptive immune system cascades and pathways, such as the TGFβ/PD-1 axis and STING-pathway [18]. Interestingly, the distinction between pure RT and adjuvant CTh is fundamental in the consideration of RT effects on the immune system. The effect of CTh on CD19+ B cells especially seems to be more profound than that of pure RT, at least in the setting in adjuvantly treated breast cancer patients [19].

The pandemic scenario of COVID-19 constituted an exception, nevertheless. Initially, the availability of the SARS-CoV-2 vaccine was limited worldwide, and vulnerable cohorts, such as the elderly and cancer patients, were prioritized for vaccination. The first vaccine licensed in Austria was BNT162b2, with a booster vaccination scheduled 21 days after an initial dose. Due to the rapid spread of the pandemic and the vaccine’s accelerated approval process, data on the vaccine’s efficacy in cancer patients were limited at the time of vaccination. To increase the safety of oncological patients at the Department of Radiation Oncology of the Ordensklinikum Linz, a monitoring program was offered to track their immune response after vaccination through the determination of bAB.

The aim of this study was to approach a holistic assessment of the immunogenicity of mRNA vaccine BNT162b2 in cancer patients, currently undergoing RT. This study investigates seroconversion, humoral immune response with antibody kinetics, bAB persistence, T-cellular response, and vaccination side effects.

## 2. Materials and Methods

### 2.1. Study Protocol

A total of 74 cancer patients receiving RT at the Department of Radiation Oncology of the Ordensklinikum Linz volunteered for weekly blood collection to determine SARS-CoV-2 bAB levels following SARS-CoV-2 vaccination. Only adult patients undergoing pure RT or combined RT with or without previous or simultaneous CTh, and who had not experienced a COVID-19 infection before, were included. The immunization schedule contained two doses of the mRNA-based BNT162b2 vaccine, administered 21 days apart, with each dose containing 0.3 mL of serum with 30 µg of the vaccine agent [20]. At least one vaccine dose was administered during RT. In the study period between January and April 2021, 46 subjects met the inclusion criteria (Figure 1). These were allocated into 2 comparative cohorts, depending on whether there was a previous or simultaneous administration of chemotherapeutical agents within the preceding month. For supplementary comparison, a group of 16 volunteering healthcare workers was enrolled as healthy controls. Participants were included if they followed the same vaccination schedule and had not experienced a COVID-19 infection before.

Demographic data of the study population, namely gender, age, ECOG-status, survival data, cancer type and specifications, cumulative radiation dose, comorbidities, and co-medications, are shown in Table 1. In all subjects, SARS-CoV-2 vaccine BNT162b2 was applied, which induces nucleoside-modified mRNA carrying lipid nanoparticles to encode the receptor binding domain (RBD) of the wild-type SARS-CoV-2 viral spike protein [21,22].

The first blood samples were taken before vaccination to assess pre-vaccination antibody status and show that there was no seroconversion before. For all patients without pre-vaccination SARS-CoV-2 antibodies against the nucleocapsid protein, five additional samples were taken in a seven-day cycle starting seven days after immunization (days 7, 14, 21, 28, and 35). In addition to SARS-CoV-2 antibodies directed against the nucleocapsid which are detectable only after SARS-CoV-2 infection, quantitative SARS-CoV-2 antibodies against the spike protein were analyzed. Patients remaining seronegative at day 35 were tested for a SARS-CoV-2-specific T-cellular reaction on the viral spike protein [23].

### 2.2. Immunogenicity Assay

Venous blood was collected in an 8 mL VACUETTE lithium heparin tube by venipuncture and processed immediately after centrifugation on the cobas e801 analyzer (Roche Diagnostics, Rotkreuz, Switzerland) according to the manufacturer’s instructions. The Elecsys^®^ Anti-SARS-CoV-2 test kit (Roche Diagnostics, Rotkreuz, Switzerland), performed to exclude a passed-through SARS-CoV-2 infection, is a double-antigen sandwich using recombinant nucleocapsid protein for the detection of total antibodies (IgA, IgM, and IgG) against SARS-CoV-2. Results are reported as numeric values in the form of a cutoff index (COI), as well as in the form of a qualitative “non-reactive” (COI < 1.0; negative) or “reactive” (COI > 1.0, positive) result [24]. To detect bAB response following vaccination, samples were tested using the Elecsys^®^ Anti-SARS-CoV-2 S kit, which quantitatively determines antibodies to the spike receptor binding domain of SARS-CoV-2 (Anti-SARS-CoV-2 S-RBD). Measurement results are expressed in U/mL, with a defined threshold of 0.80 U/mL to distinguish samples as “reactive” (≥0.80, positive) and “non-reactive” (<0.80 U/mL, negative) for SARS-CoV-2 RBD-specific antibodies. Both antibody tests measure predominantly, but not exclusively, IgG, which is considered a long-lasting reliable marker of humoral response and may be detectable for months to years after SARS-CoV-2 exposure [16].

### 2.3. SARS-CoV-2-Specific T-Cell Assay

Patients with no bAB seroconversion were offered to undergo an analysis of SARS-CoV-2-specific T-cells via a highly specific SARS-CoV-2-derived T-cell epitope analysis, which provides information about a previous antigen contact, causing a stimulation of the adaptive immune response [25]. T-cell immunity to the SARS-CoV-2 Spike protein was assessed in seronegative patients using an IFN-gamma release assay.

### 2.4. Statistical Analysis

Data were analyzed descriptively. bAB levels were plotted for each sampling day. Box plots were generated to show the variation in the data. bAB levels were plotted as log10. Due to a clear deviation from the requirements for the use of parametric analysis methods (normal distribution and homogeneity of variance), non-parametric methods were used for analysis. The Mann–Whitney U test was used for differences between cohorts, and the Wilcoxon signed-rank test was used for differences between time points. As the aim of this study was to determine the development of bAB over time as a function of the three cohorts, *p*-values are presented only to identify interesting differences. The underlying significance tests are in no way intended to test hypotheses. Therefore, no alpha error correction for multiple testing is applied. Analyses were performed using R version 4.2.2 or SPSS version 28.

### 2.5. Ethics

The study protocol received approval (number: 1214/2021) from the ethics committee of Upper Austria (Medical Faculty, Johannes Kepler University in Linz, Austria) in accordance with the Declaration of Helsinki.

## 3. Results

Between January and April 2021, 46 patients treated with RT plus/minus an additional CTh were entered onto the study. The two experimental samples were complemented with a third cohort of healthy controls. The following aspects were investigated:

### 3.1. Patient Characteristics

Cohort 1 contains 23 patients, treated with pure RT alone. Cohort 2 contains 23 patients, treated with CTh previous or simultaneous to RT. Cohort 3 contains 16 healthy control subjects. Additional characteristics, including age, ECOC status, the survival rate after 2.5 years, and mean survival, are reported in Table 1. Furthermore, oncologic entities specified into stage, treatment intention, and the mean cumulative dose of irradiation, as well as comorbidities and concomitant medication, were captured. In cohorts 1 and 2, the most frequently occurring malignancies were prostate, breast, and lung cancer. In cohort 1, there were 15 (65.2%) patients with prostate cancer. There was a high prevalence of arterial hypertension in both cohorts 1 and 2. In cohort 2, three (13.0%) subjects and in cohort 1, one (4.4%) subject, respectively, were under steroid treatment while obtaining RT. The details of the comorbidities and co-medication are provided in the Appendix A Table A1.

### 3.2. Seroconversion

Qualitative bAB measurements were performed at six time points and provide seroconversion values. While 100% of the individuals in the control cohort already had seroconversion on day 14, the other two cohorts lagged behind, especially those patients who received vaccination during RT plus CTh. In the final measurements on day 35, cohort 1 shows comparable seropositivity rates (95.3%) to cohort 3 (100%), while cohort 2 (77.8%) differs. In cohort 1, one (4.7%) subject remained seronegative; in cohort 2, four (22.2%) subjects remained seronegative; and in cohort 3, there was no seronegative subject. The details are shown in Figure 2.

### 3.3. Humoral Immune Response

Quantitative bAB titer values are provided in Table 2. SARS-CoV-2 nucleocapsid bAB remained negative in each measurement. The significance of titer value differences between each cohort was calculated. On measurement day 0, the test was not applicated. On day 7, cohort 1 had no significant difference to cohort 2 (z = −1.564, *p* = 1.000) and cohort 3 (z = −1.437, *p* = 0.151). From day 14 onwards, cohort 2 had significantly lower values compared to cohort 3, as well as cohort 1. In the final measurements on day 35, cohort 1 had significantly higher bAB values than group 2 (z = −3.030, *p* = 0.002) and comparable values to group 3 (z = −1.402, *p* = 0.297), which had again significantly higher bAB values than group 2 (z = −4.798, *p* < 0.001).

### 3.4. Kinetics

Between day 0 and day 7, the assessment of kinetics was not applicable. Between day 7 and 14, there was a significant bAB increase in cohort 1 (z = −2.201, *p* = 0.028) and a highly significant in cohort 3 (z = −3.516, *p* < 0.001). In cohort 2, the increase was insignificant (z = −1.826, *p* = 0.068). Between day 14 and 21, there was a significant increase in cohorts 1 (z = −3.059, *p* = 0.002), 2 (z = −2.197, *p* = 0.028), and 3 (z = −3.258, *p* = 0.001). Between day 21 and 28, there was a significant increase in cohorts 1 (z = −2.934, *p* = 0.003), 2 (z = −1.997, *p* < 0.001), and 3 (z = −3.516, *p* < 0.001). Between day 28 and 35, there was an insignificant increase in cohort 1 (z = −0.764, *p* = 0.445), whereas in cohort 2 (z = −2.207, *p* = 0.027) and 3 (z = −2.585, *p* = 0.010), the increase was significant. Additionally, the kinetics of bAB formation within the interval between each vaccination administration are depicted in Figure 3.

### 3.5. SARS-CoV-2-Specific T-Cell Measurements

In four of the five patients in our collective without seroconversion, SARS-CoV-2-specific T cells were measured at a median of 8 weeks after the second vaccine dose. In all four patients, no SARS-CoV-2-specific T cells were detected in the T-cell assay.

### 3.6. Third Vaccine Dose

Within an individualized treatment concept, a third vaccine dose was offered to patients with persistently negative bAB and no evidence of SARS-CoV-2-specific T cells. Four patients received a third dose at a median of 21 weeks (8–29 weeks) after the second BNT162b2 dose. In all four patients, positive bAB was subsequently detected within 35 days after application of the third dose.

### 3.7. Vaccination Side Effects

Side effects were assessed in 15 patients of cohort 1 and 13 patients of cohort 2 after each administration separately and are provided in Table 1. The initial dose was better tolerated in cohort 2 than in cohort 1, while the booster dose was better tolerated in cohort 1 than in cohort 2. Cohort 3 had similar frequencies of side effects as cohort 1 after each administration. No serious side effects were observed in any cohort. More detailed information about the side effects is provided in the Appendix A Table A1.

### 3.8. Binding Antibody Persistence

Long-term measurements at about a median of 5–7 months after the first vaccine dose were performed in 9 patients of cohort 1, in 8 patients of cohort 2, and 16 healthy subjects of cohort 3. Long-term measurements were only analyzed in those patients who were seropositive at day 35 after the first vaccination. There was a persistent seropositivity of 100% in all cohorts (Table 2). A seroconversion of SARS-CoV-2 nucleocapsid antibodies was detected in none of the patients. Cohorts 1 and 3 had comparable quantitative bAB titers, while the titer values were lower in cohort 2.

## 4. Discussion

To the best of our knowledge, these are the first published data investigating serial longitudinal immunogenicity parameters of SARS-CoV-2 immunization in cancer patients receiving RT. Our findings show that patients receiving pure RT elicit similar, although somehow retarded, binding antibody kinetics as healthy controls. In patients with sequential or concomitant RT and CTh, however, humoral response is impaired with a markedly postponed incline of antibodies and substantially lower titers on day 35 after the first vaccination. This finding has an impact on counselling patients regarding immunization during pure RT and during RT plus CTh, respectively. Additionally, regarding aspects of the interplay between the immune system and human cancers, as well as its therapy, as mentioned in the introduction, RT might possess the ability to interfere with a vaccine-induced immune response. Especially, the irradiation of the bone marrow, which contains substantial precursor B cells for antigen presentation is crucial and could cause immunosuppressive side effects, analogous to CTh. Additionally, mRNA immunization uniquely enables a systemic delivery of antigens in contrast to other vaccine platforms [26]. Therefore, local interferences are likely to be well tolerated. However, there are limited works in the literature referring to the vaccination of cancer patients while obtaining radiotherapy. Related research investigates only isolated parameters such as seroconversion, bAB titer, or side effects in a separate way. There is no published comprehensive analysis of serial seroconversion, HIR, and kinetics in cancer patients under RT within the first 35 days after the initial vaccination, nor is there any direct comparison of these parameters to those of a healthy control cohort vaccinated within the same period published so far. The details of various vaccine-induced immunogenicity parameters are discussed below.

**Seroconversion**: Our data disclose that pure RT does not seem to compromise qualitative seroconversion. Patients in cohort 1 had a seropositivity rate of 95.2% after 35 days. The healthy control cohort had a comparable rate of 100%. On the other hand, we observed a substantially decreased seroconversion rate of 77.8% in cohort 2 (RT plus CTh). This corresponds to the findings of surveys on the seroconversion in cancer patients, following SARS-CoV-2 vaccination in temporal proximity to RT, who underwent onetime measurements. After two doses, there were seropositivity rates of 95% when immunized by a vector-based vaccine [27] and of 93.5% in those immunized by an mRNA vaccine [28]. There are no further studies directly comparing cohorts of pure RT and combined RT/CTh.

**Non-responders**: A total of five seronegative subjects were observed at day 35. Of these, four were on a corticosteroid therapy while they were vaccinated. In two cases, corticosteroids were part of a chemotherapy regimen, while in the remaining two, they were given for other indications. The remaining patient had a reduced general condition with an ECOG-status of 3. In line with the literature, the corticosteroid intake and reduced general condition are known to be related to reduced immune responses [29]. The bAB titer was close below the positivity threshold in one of the five patients and not detectable in the remaining four.

**SARS-CoV-2-specific T cells:** In four of five seronegative patients, a SARS-CoV-2-specific T-cell assay was conducted. No evidence of SARS-CoV-2-specific T cells was disclosed in any of these four patients. Thus, in these cases, negative bAB also reflected a negative T-cell response. Increased odds of a mismatch between serological and T-cellular response in cancer patients after SARS-CoV-2 vaccination have been reported previously [30]. An absent seroconversion after COVID-19 mRNA vaccination at a simultaneously positive T-cell response was observed in 24–74% of hematologic [31,32] and 26% of solid cancer patients [31].

**Third vaccine dose**: Patients, in which we could not detect a specific antibody response, were offered, as an individualized concept, a third immunization with BNT162b2. Additionally, recipients of a third vaccination dose had to be negative in the SARS-CoV-2-specific T-cell assay. Currently, multiple SARS-CoV-2 boost vaccinations are administrated regularly and highly recommended, especially for cancer patients. [33], but in the middle of 2021, this was an off-label use, as there was limited evidence for a third dose. The theory behind the application of another vaccine dose was that the patient’s cancer treatment during the vaccination might have prohibited a sufficient immune response. This response, however, might arise when an additional, second booster vaccine dose was given after the immunocompromising effect of cancer treatment had levelled off. In fact, all seronegative patients who received a third vaccination dose had seroconversion afterwards. Even one subject who was still under steroid treatment while receiving the third vaccine showed positive seroconversion afterwards. Although only practiced in a few patients, our data show that this tactic might be applicable in cancer patients in future epidemic scenarios: those vaccinated during active tumor therapy could be monitored for HIR and would receive a booster dose after the end of active treatment in case of a non-response. Moreover, there are several reports of a high effectiveness of further booster administrations in cancer patients [34,35]

**Antibody titers**: Our data disclose that pure RT does not compromise quantitative HIR after SARS-CoV-2 vaccination. The median and mean bAB titer values of pure RT patients were comparable to healthy controls on day 35 after the first vaccination, while patients receiving combined CTh and RT achieve substantially diminished values. We derive a recommendation for the vaccination of cancer patients while obtaining pure RT, especially in situations with high infection rates and/or a moderate-to-high likelihood of severe infection outcomes.

With regard to pure CTh, there is a body of published evidence that bAB formation after vaccination with BNT162b2 vaccine is delayed or diminished [32,33,36,37,38]. Therefore, and in combination with our findings, we suspect that CTh is the major reason for the observed reduced HIR in patients vaccinated during RT in conjunction with CTh.

Apart from oncologic therapies, there are patient-, disease-, and vaccine-related factors that influence bAB titers after vaccination in oncologic patients [39,40]. With respect to the patient’s general condition, to the ratio of lung cancer patients, to the extent of concomitant disease, and to the distribution of concomitant medication, cohort 1 and cohort 2 were comparable (Table 1). Other confounding factors, like previous SARS-CoV-2 infection, type of vaccine, and interval between vaccinations, were primarily controlled for by selecting only nucleocapsid negative patients who were receiving two doses of BNT 162b2 three weeks apart.

**Kinetics**: Overall kinetics in healthy controls and pure RT patients were comparable. RT + CTh patients demonstrated more sluggish kinetics and lower bAB titers at all timepoints beyond day 7. Cancer patients tend to have delayed and diminished HIRs following SARS-CoV-2 immunization, especially in the context of CTh [30]. There are no published data on bAB kinetics of cancer patients undergoing RT. Particularly, we want to stress the role of the boost dose (second vaccination) in the context of kinetics. There was a tendency towards a slightly decreased HIR in pure RT patients after the first vaccination, which was fully compensated after the second vaccine administration (Figure 3).

**Antibody longevity**: At long-term follow-up (median of 5–7 months after the first vaccination), patients with seroconversion were tested for antibodies again. All tested patients still had measurable antibodies, although at reduced levels as compared to day 35. No nucleocapsid seroconversion was detected in patients with long-term measurements, indicating that they experienced no SARS-CoV-2 infection within this time frame. Pure RT patients displayed substantially higher persistent titers compared to patients that had received additional CTh. Consistent with our findings, it has been shown by others that BNT162b2 ensures long-term persist protection and has high efficacy rates against the predominant SARS-CoV-2 variants of 2021 [41]. One study in healthy subjects disclosed a high rate of positive bAB titers (99.8%) 3.4 months after initial vaccination, with median bAB titers of 990 U/mL [42]. This fits to the titer levels of 646 U/mL, which we observed at a median of 7 months after the first vaccination in healthy controls.

**Adverse vaccination effects:** In our investigation, the vaccine was well tolerated in all three cohorts, and only mild or moderate side effects occurred. No severe vaccination reactions were observed. The more pronounced immune response after the first vaccination dose in cohort 1 (as compared to cohort 2) was preceded by more vaccination side effects in this group. On the other hand, the higher rise of bAB in cohort 2 after the second vaccine dose went along with more side effects of the boost dose in this group. Interestingly, all patients with pyrexia had bAB titers far beyond the mean and medium values. Individuals in cohort 3 exhibited a similar side effect pattern to those in cohort 1, and this was followed by similar humoral immune responses and antibody kinetics. This is congruent to published data showing that adverse vaccination reactions indicate more efficient immune responses and higher levels of bAB titers [43,44]. A higher level of bAB, on the other hand, is associated with better protection against COVID-19 [7,45].

## 5. Conclusions

Cancer patients undergoing pure RT during the time of SARS-CoV-2 mRNA vaccination showed a proper immune response: seroconversion rates, binding antibody titers at day 35, and side effects were comparable to healthy controls. Patients receiving CTh in conjunction with RT demonstrated an impaired HIR as compared to healthy controls and to those receiving pure RT within 35 days after immunization. We highly suggest vaccination of oncologic patients undergoing radiotherapy in future epidemic or pandemic scenarios, at least when mRNA vaccines are applied. However, in those receiving RT in conjunction with CTh, humoral response should be monitored after vaccination, as non-responders might benefit from additional vaccine doses after the end of antineoplastic therapy.

## 6. Limitations

This study has some limitations owing to its small sample size and retrospective design. Not all patients adhered to the schedule of weekly measurements, though this must be judged against the background of a rampant pandemic. Nevertheless, the data provide new insights into vaccine efficiency and tolerability in oncologic patients undergoing radiotherapy ± chemotherapy.

## Figures and Tables

**Figure 1 vaccines-12-00275-f001:**
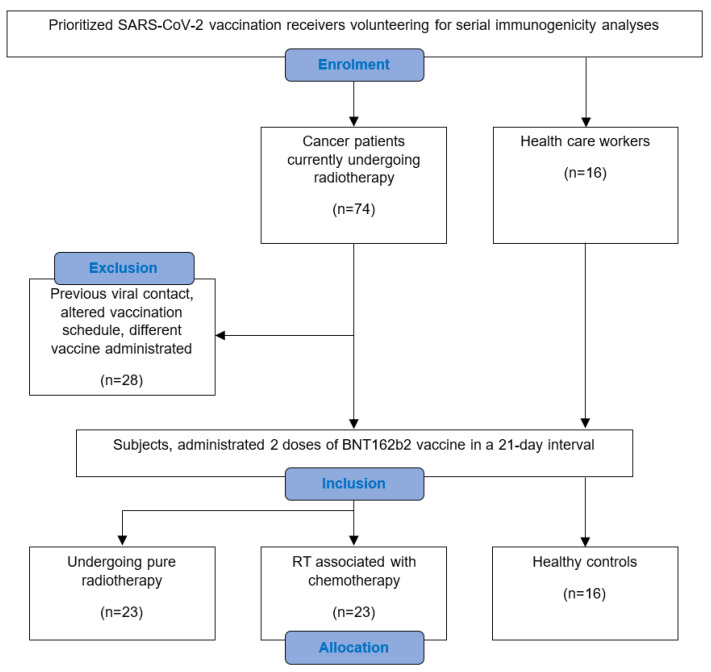
Consort flow diagram of the study conception in detail.

**Figure 2 vaccines-12-00275-f002:**
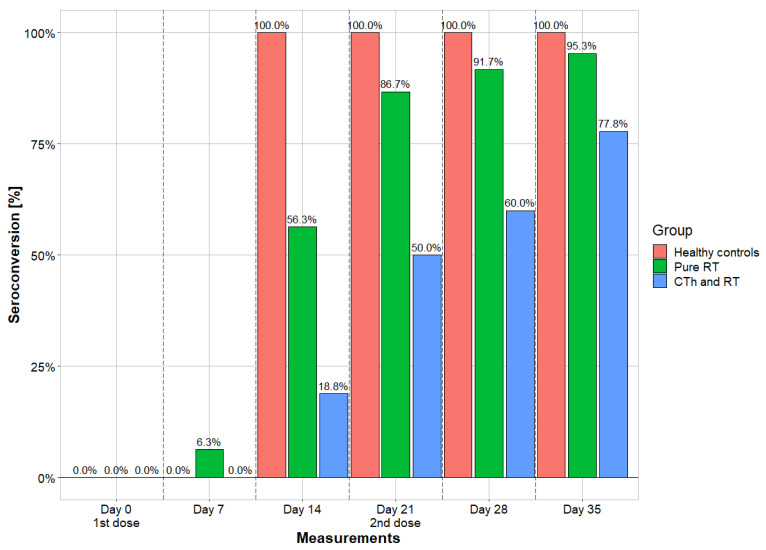
Seroconversion rates of the 3 cohorts. Measurements were taken in a seven-day cycle and portrayed in comparing longitudinal fashion. Vaccines were administrated on day 0 and day 21. RT = radiotherapy; CTh = chemotherapy.

**Figure 3 vaccines-12-00275-f003:**
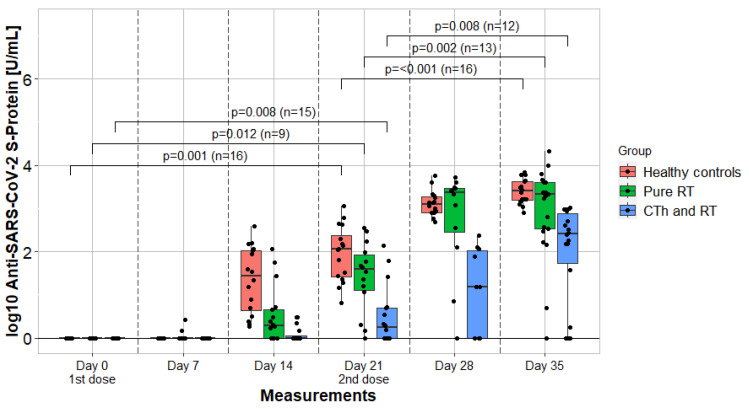
Humoral immune response and kinetics of 3 cohorts. Measurements were taken in a seven-day cycle and portrayed by comparing in a longitudinal fashion. Vaccines were administrated on day 0 and day 21. The *p*-values compare binding antibody titer values after each vaccination. RTh = radiotherapy; CTh = chemotherapy.

**Table 1 vaccines-12-00275-t001:** Baseline demographics and data of all 3 study cohorts.

Variables	Cohort 1 RT Alone	Cohort 2 RT and CTh	Cohort 3 Healthy Controls
**Sex**			
Female	6 (26.1)	9 (39.1)	5 (31.3)
Male	17 (73.9)	14 (60.9)	11 (68.6)
**Age (years)**			
Min	39	50	27
Max	92	81	70
Mean (SD)	72.0 (11.3)	66.6 (9.7)	45.9 (11.8)
**ECOG**			
0	13	12	16
1	9	8	0
2	1	2	0
3	0	1	0
4	0	0	0
**Survival**			
Alive	17 (73.9)	17 (73.9)	16 (100)
Deceased	6 (26.1)	6 (26.1)	0 (0)
**Cancer entity**			n.a.
Prostate	15	3	
Mean radiation dose	57.4 Gy	49.3 Gy	
Breast	2	5	
Mean radiation dose	47.0 Gy	37.8 Gy	
Lung	3	4	
Mean radiation dose	42.5 Gy	45 Gy	
Brain	1	3	
Mean radiation dose	60 Gy	60 Gy	
Skin (squamous cell)	2	3	
Mean radiation dose	57.5 Gy	62.1 Gy	
Esophagus	0	2	
Mean radiation dose		47.0 Gy	
Rectal	0	2	
Mean radiation dose		7.5 Gy	
Pancreas	0	1	
Mean radiation dose		54 Gy	
**Stage**			n.a.
Localized	18 (78.3)	20 (87)	
Metastatic disease	5 (21.7)	3 (13)	
**Treatment intention**			n.a.
Curative	18 (78.3)	20 (87)	
Palliative	5 (21.7)	3 (13)	

Numbers are No.; (%), numeral impairments due to round errors; n.a. = not applicable.

**Table 2 vaccines-12-00275-t002:** Descriptive statistical titer values of humoral response due to SARS-CoV-2 immunization.

Humoral Response of Antibodies (U/mL)	Cohort 1 Pure RT	Cohort 2 CTh and RT	Cohort 3 Healthy Controls
**n day 0**	23	23	16
Min	0	0	0
Max	0	0	0
Mean (SD)	n.a.	n.a.	n.a.
Median (IQR)	n.a.	n.a.	n.a.
**n day 7**	16	19	16
Min	0	0	0
Max	1.60	0	0
Mean (SD)	0.13 (0.40)	0 (0)	0 (0)
Median (IQR)	0 (0–0)	0 (0–0)	0 (0–0)
**n day 14**	16	16	16
Min	0	0	0.86
Max	112	2.06	381
Mean (SD)	12.95 (29.25)	0.36 (0.70)	69.30 (96.46)
Median (IQR)	0.91 (0–3.61)	0 (0–0.12)	26.85 (3.46–104.50)
**n day 21**	14	16	16
Min	0	0	5.35
Max	358	138	1129
Mean (SD)	82.22 (110.29)	14.75 (35.21)	217.03 (291.63)
Median (IQR)	38 (11.78–85.35)	0.75 (0–3.95)	112.5 (24.73–249.50)
**n day 28**	12	10	16
Min	0	0	437
Max	5161	237	5747
Mean (SD)	2001.34 (1615.49)	57.76 (75.57)	1640.06 (1351.12)
Median (IQR)	2334.5 (299.25–2962.5)	14.10 (0–104.1)	1236 (806.75–1834)
**n day 35**	21	18	16
Min	0	0	804
Max	20,698	1055	6716
Mean (SD)	3165.71 (4588.48)	395.33 (382.82)	2994.13 (1855.27)
Median (IQR)	2135 (338–3920)	285.50 (64.8–783.75)	2509 (1562–4253.25)
**n long term persistency**	9	8	16
Min	64.2	65.7	55
Max	2017	869	1936
Mean (SD)	587.5 (563.7)	267.5 (242.6)	694.4 (408.0)
Median (IQR)	481.5 (344.25–528.75)	188 (120–265.75)	646 (441.25–836.5)
**Days after d35**			
Mean (SD)	1203.9 (31.2)	152.13 (56.5)	196.4 (44.6)
Median (IQR)	185 (182–202)	131 (50.5–170)	174 (149.5–180.5)

Binding antibody titers in U/mL; n.a. = not applicable; SD = standard deviation; IQR = interquartile range; n = number of subjects tested on a particular measurement day.

## Data Availability

Data directly supporting this investigation can be obtained by contacting the corresponding author.

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
