# Peer review of "Immunogenicity Parameters of Cancer Patients Receiving the mRNA Vaccine BNT162b2 While Obtaining Radiotherapy: A Longitudinal Cohort Evaluation"

_vaccines, 2024, doi:10.3390/vaccines12030275_

Round 1
Reviewer 1 Report
Comments and Suggestions for Authors
The authors have approached an important subject.
Purpose
“As the aim of this study is to determine the development of bAB over time as a function of the three cohorts, p-values are presented only to identify interesting differences. The underlying significance tests are in no way intended to test hypotheses. Therefore, no alpha error correction for multiple testing is applied.”
Design
"The collective was complemented with a third cohort of healthy controls.” [It is clear ”the collective” means the two experimental samples but perhaps say so. I have never seen the term “collective“ before].
The samples are tiny: 6/23, 9/23 and 5/16 female in cohorts 1, 2 and healthy controls; 18 prostate, 7 breast and 7 lung cancers; 18/23 and 2/23 localised cancer; most common comorbidities HTN 10/23 and 11/23
Results
The authors are very clear that they are not testing hypotheses.
Nevertheless, the authors presented a detailed statistical analysis and “Due to a clear deviation from the requirements for the use of parametric analysis methods (normal distribution and homogeneity of variance), non-parametric methods were used for analysis.”
In Figure 2 the box plots contain tiny numbers within the squares.
Conclusions
“Cancer patients, undergoing pure RT during the time of SARS-CoV-2 vaccination showed a proper immune response: seroconversion rates, binding antibody titres at day 35 and side effects were comparable to healthy controls. Patients receiving CTh in conjunction with RT demonstrated an impaired HIR as compared to healthy controls and to those receiving pure RT. We highly suggest vaccination of oncologic patients undergoing radiotherapy in future epidemic or pandemic scenarios. However, in those receiving RT in conjunction with CTh, humoral response should be monitored after vaccination as non-responders might benefit from additional vaccine doses after the end of antineoplastic therapy.”
“This study has some limitations owing to its small sample size and retrospective design. Not all patients adhered to the schedule of weekly measurements, though this must be judged against the background of a rampant pandemic. Nevertheless, the data provide new insights in vaccine efficiency and tolerability in oncologic patients undergoing radiotherapy +/- chemotherapy.”
I agree with the authors' self assessed major limitations.
The authors approached an important subject. It is likely that physicians would protect such immune compromised patients with appropriate vaccines.
I think this article demonstrates appropriate methods to research the subject and perhaps would best be shortened into a research note. The extensive tables are inappropriate with this tiny sample.
English text is impeccable.
Author Response
Dear Reviewer,
we hope this letter finds you in the best spirits!
Thank you for giving us the opportunity to submit a revised draft of our manuscript Immunogenicity parameters of cancer patients receiving the mRNA vaccine BNT162b2 while obtaining radiotherapy: a longitudinal cohort evaluation. We appreciate the time and effort for your reviewing and the dedication of your accurate and valuable feedback to enhance the quality of our work. We have been able to incorporate changes to reflect most of the provided suggestions. The changes within the manuscript are highlighted.
Here is a point-by-point response to your comments and concerns:
The authors have approached an important subject.
Purpose
“As the aim of this study is to determine the development of bAB over time as a function of the three cohorts, p-values are presented only to identify interesting differences. The underlying significance tests are in no way intended to test hypotheses. Therefore, no alpha error correction for multiple testing is applied.”
Design
"The collective was complemented with a third cohort of healthy controls.” [It is clear ”the collective” means the two experimental samples but perhaps say so. I have never seen the term “collective“ before].
We kindly thank you for the advice, the pertaining term was replaced.
The samples are tiny: 6/23, 9/23 and 5/16 female in cohorts 1, 2 and healthy controls; 18 prostate, 7 breast and 7 lung cancers; 18/23 and 2/23 localised cancer; most common comorbidities HTN 10/23 and 11/23
As mentioned in the limitations section, the small sample size is suboptimal and explained by the selected group of potential subjects to include.
Results
The authors are very clear that they are not testing hypotheses.
Nevertheless, the authors presented a detailed statistical analysis and “Due to a clear deviation from the requirements for the use of parametric analysis methods (normal distribution and homogeneity of variance), non-parametric methods were used for analysis.”
In Figure 2 the box plots contain tiny numbers within the squares.
We kindly thank you for the advice, the pertaining figure was re-edited.
Conclusions
“Cancer patients, undergoing pure RT during the time of SARS-CoV-2 vaccination showed a proper immune response: seroconversion rates, binding antibody titres at day 35 and side effects were comparable to healthy controls. Patients receiving CTh in conjunction with RT demonstrated an impaired HIR as compared to healthy controls and to those receiving pure RT. We highly suggest vaccination of oncologic patients undergoing radiotherapy in future epidemic or pandemic scenarios. However, in those receiving RT in conjunction with CTh, humoral response should be monitored after vaccination as non-responders might benefit from additional vaccine doses after the end of antineoplastic therapy.”
“This study has some limitations owing to its small sample size and retrospective design. Not all patients adhered to the schedule of weekly measurements, though this must be judged against the background of a rampant pandemic. Nevertheless, the data provide new insights in vaccine efficiency and tolerability in oncologic patients undergoing radiotherapy +/- chemotherapy.”
I agree with the authors' self assessed major limitations.
The authors approached an important subject. It is likely that physicians would protect such immune compromised patients with appropriate vaccines.
I think this article demonstrates appropriate methods to research the subject and perhaps would best be shortened into a research note. The extensive tables are inappropriate with this tiny sample.
In line with your suggestions, the text was strengthened and the tables were abridged
English text is impeccable.
We thank you for your correspondence, providing a detailed feedback and the improvement issues to enhance the quality of this manuscript.
Additional clarifications were done all over the manuscript and highlighted.
We look forward to hearing from you in due time regarding our submission and to respond to any further questions and comments you may have.
Sincerely,
Paul Thöne

Reviewer 2 Report
Comments and Suggestions for Authors
Review Report for authors:
The manuscript entitled “Immunogenicity parameters of cancer patients receiving the mRNA vaccine BNT162b2 while obtaining radiotherapy: a longitudinal cohort evaluation”, is an interesting piece of research presenting a limited measurements of the immune response to mRNA based COVID vaccine in cancer patients that received different types of radiotherapy. Interestingly the results of the study show that radiotherapy alone did not interfere with humoral response development, unless combined with chemotherapy. Nevertheless, data were not statistically processed to show significant differences but rather the authors claimed that: “The underlying significance tests are in no way intended to test hypotheses.”, which is more or less to accept that the observed differences could be due to chance or that the differences could actually exist (but we do not know if these are real).
In order to be accepted for publication the authors should incorporate the following corrections:
1) Process the data by more than one statistically available method, assuming:
a) Treat as binary data: a cutt-off value to dichotomize the data in responders/non responders or positive /negative to the presence of antibodies.
b) Treat titers as a continuos variable: having the titers from the various analyses performed, find differences among the ranks obtained by any non-parametric test. A valid hypothesis is that RT+CTh results in antibody titers significantly different than RT alone and then corroborate no difference between RT and controls (you can use a T test for it if it is normally distributed, if not, you can use a non parametric for just two samples, since these are the only two comparisons needed – and biologically relevant-).
2)In line 20 to 21 of the Abstract you wrote: “necessitating excep
tional protective measures, such as vaccination”... As it is written it seems that vaccination is an extraordinary measure, please rephrase.
3) In lines 21 to 24 of the Abstract you wrote: “there is no specific information about vaccination immunogenicity.”……...“serial antibody measurements were therefore offered to cancer patients, following SARS-CoV-2 vaccination while obtaining radiotherapy.” You wrote this in reference to a measurement of immunity, and it is just “humoral immunity” what you are measuring. Please talk just about humoral immunity and not “vaccination immunogenicity”.
4) I saw your Inclusion criteria and I wonder... Why you did not use the people receiving a different vaccine type as a control? Specially because this mRNA vaccine is not a classical vaccine platform and it is the 1st time it is used in humans. Moreover, the immunogenicity is quite different since this mRNA vaccines has been reported to produce spike protein not only at the site of injection but also systemically in a non-controlled, and profuse manner. Meanwhile, most classical vaccine platforms release target antigens only at the injection site, which also is connected to the concept of “draining lymph node” which apparently wouldn’t be possible to conceive in this mRNA vaccines. Having this in mind:
a) I would like you to include this unique features of mRNA vaccines in the discussion, as a potential cause of the lack of significant differences between patients receiving RT and controls.
b) Is there any possibility to include results from cancer patients that received other type of COVID vaccines?
5) In lines 101, if you wrote Figure 1, it is redundant to write that this is a Figure.
6) In line 110 the period goes after the reference.
7) Table 1 is too long (it takes almost 3 pages). Could you please present part of the data as graphs/diagrams, to make it more easy to go trough and eye catching also?
8)Include in the Figure 2 some marks indicating days for vaccination and boosts. You can start from 0 (zero) or in a minus 5 (-5) days your timeline (if desired) to better visualize the first dose.
9) In line 222: Please do not sub-divide the Discussion section, and discuss the results without a background introduction, the background actually goes in the introduction section, unless it is explanatory of the observed results, like for example: the nature of the vaccine, the kind or RT given (see point 10), etc.
10) In lines 256-257 of the Discussion you wrote: “ ...RT might possess the ability to interfere with a vaccine-induced immune response,... “ I suggest that rather to discuss this in general, you may want to focus in the type of radiation therapy applied to the patient, since it won't have the same effect a focused radiation for prostatic cancer than any other type. You can also consider that B cell precursors in bone marrow (BM) should be protected from most types of radiation, and thus it can explain why RT had no negative effect on HIR. On the other hand, CTH would reach BM, affecting the normal development of B Lymphocytes precursors and its associated HIR. You can add this to the manuscript.
11) In Conclusions, line 359 please clarify again that this is the mRNA based SARS-CoV vaccination and not other.
12) In line 362 please clarify that the difference is only at early times after immunization or to 1st dose.
13) In lines 363-364 you wrote: “ We highly suggest vaccination of oncologic patients undergoing radiotherapy in future epidemic or pandemic scenarios”. Vaccination is usually carried out by employing classical vaccines and not mRNA vaccines, your results, pitifully are restricted to COVID mRNA vaccines, and particularly only to BNT162b2 vaccine. To generalize you need at least a control group receiving a vaccine based on a classical platform. Please rephrase.
If all the corrections are incorporated, the statistic is revised and the presentation is improved the paper could be accepted for its publication in Vaccines.
Best Regards,
The reviewer.
Author Response
Dear Reviewer,
we hope this letter finds you in the best spirits!
Thank you for giving us the opportunity to submit a revised draft of our manuscript Immunogenicity parameters of cancer patients receiving the mRNA vaccine BNT162b2 while obtaining radiotherapy: a longitudinal cohort evaluation. We appreciate the time and effort for your reviewing and the dedication of your accurate and valuable feedback to enhance the quality of our work. We have been able to incorporate changes to reflect most of the provided suggestions. The changes within the manuscript are highlighted.
Here is a point-by-point response to your comments and concerns:
The manuscript entitled “Immunogenicity parameters of cancer patients receiving the mRNA vaccine BNT162b2 while obtaining radiotherapy: a longitudinal cohort evaluation”, is an interesting piece of research presenting a limited measurements of the immune response to mRNA based COVID vaccine in cancer patients that received different types of radiotherapy. Interestingly the results of the study show that radiotherapy alone did not interfere with humoral response development, unless combined with chemotherapy. Nevertheless, data were not statistically processed to show significant differences but rather the authors claimed that: “The underlying significance tests are in no way intended to test hypotheses.”, which is more or less to accept that the observed differences could be due to chance or that the differences could actually exist (but we do not know if these are real).
In order to be accepted for publication the authors should incorporate the following corrections:
1) Process the data by more than one statistically available method, assuming:
- a) Treat as binary data: a cutt-off value to dichotomize the data in responders/non responders or positive /negative to the presence of antibodies.
- b) Treat titers as a continuos variable: having the titers from the various analyses performed, find differences among the ranks obtained by any non-parametric test. A valid hypothesis is that RT+CTh results in antibody titers significantly different than RT alone and then corroborate no difference between RT and controls (you can use a T test for it if it is normally distributed, if not, you can use a non parametric for just two samples, since these are the only two comparisons needed – and biologically relevant-).
We kindly thank you for suggesting this statistical issues. Due to the small number of cases and the existing outliers, parametric methods were generally not used. As the results are significant despite the less sensitive non-parametric methods used, an additional evaluation with t-tests or variance analyses would not provide any further insight (especially as the data are not normally distributed due to the outliers mentioned and their use would be viewed critically anyway).
2)In line 20 to 21 of the Abstract you wrote: “necessitating exceptional protective measures, such as vaccination”... As it is written it seems that vaccination is an extraordinary measure, please rephrase.
The pertaining passages were rephrased.
3) In lines 21 to 24 of the Abstract you wrote: “there is no specific information about vaccination immunogenicity.”……...“serial antibody measurements were therefore offered to cancer patients, following SARS-CoV-2 vaccination while obtaining radiotherapy.” You wrote this in reference to a measurement of immunity, and it is just “humoral immunity” what you are measuring. Please talk just about humoral immunity and not “vaccination immunogenicity”.
The pertaining passage was rephrased, following your suggestion
4) I saw your Inclusion criteria and I wonder... Why you did not use the people receiving a different vaccine type as a control? Specially because this mRNA vaccine is not a classical vaccine platform and it is the 1st time it is used in humans. Moreover, the immunogenicity is quite different since this mRNA vaccines has been reported to produce spike protein not only at the site of injection but also systemically in a non-controlled, and profuse manner. Meanwhile, most classical vaccine platforms release target antigens only at the injection site, which also is connected to the concept of “draining lymph node” which apparently wouldn’t be possible to conceive in this mRNA vaccines. Having this in mind:
- a) I would like you to include this unique features of mRNA vaccines in the discussion, as a potential cause of the lack of significant differences between patients receiving RT and controls.
Thank you for this valuable suggestion, which is now added into the discussion section.
- b) Is there any possibility to include results from cancer patients that received other type of COVID vaccines?
Unfortunately, there were only few patients having received different types of vaccines, as the federal vaccination schedule of Austria intended cancer patients to follow BNT162b2. Variations only occurred in exceptional cases.
5) In lines 101, if you wrote Figure 1, it is redundant to write that this is a Figure.
This was a formation error. The proper sentence was inserted.
6) In line 110 the period goes after the reference.
The reference is now adjusted
7) Table 1 is too long (it takes almost 3 pages). Could you please present part of the data as graphs/diagrams, to make it more easy to go trough and eye catching also?
The table was shortened and an appendix was created.
8)Include in the Figure 2 some marks indicating days for vaccination and boosts. You can start from 0 (zero) or in a minus 5 (-5) days your timeline (if desired) to better visualize the first dose.
The pertaining figure was optimized following your suggestions.
9) In line 222: Please do not sub-divide the Discussion section, and discuss the results without a background introduction, the background actually goes in the introduction section, unless it is explanatory of the observed results, like for example: the nature of the vaccine, the kind or RT given (see point 10), etc.
Following your suggestions, the introductive paragraphs of the discussion section were modified and presented in the introduction section and the sub-division was not overtaken. The first introductive word of the remaining paragraphs were left to provide an accurate overview and keep up the format of the results section.
10) In lines 256-257 of the Discussion you wrote: “ ...RT might possess the ability to interfere with a vaccine-induced immune response,... “ I suggest that rather to discuss this in general, you may want to focus in the type of radiation therapy applied to the patient, since it won't have the same effect a focused radiation for prostatic cancer than any other type. You can also consider that B cell precursors in bone marrow (BM) should be protected from most types of radiation, and thus it can explain why RT had no negative effect on HIR. On the other hand, CTH would reach BM, affecting the normal development of B Lymphocytes precursors and its associated HIR. You can add this to the manuscript.
We thank you for this valuable content and added a passage into the discussion section.
11) In Conclusions, line 359 please clarify again that this is the mRNA based SARS-CoV vaccination and not other.
The clarification was specified
12) In line 362 please clarify that the difference is only at early times after immunization or to 1st dose.
Following your suggestion, an additional specification was inserted
13) In lines 363-364 you wrote: “ We highly suggest vaccination of oncologic patients undergoing radiotherapy in future epidemic or pandemic scenarios”. Vaccination is usually carried out by employing classical vaccines and not mRNA vaccines, your results, pitifully are restricted to COVID mRNA vaccines, and particularly only to BNT162b2 vaccine. To generalize you need at least a control group receiving a vaccine based on a classical platform. Please rephrase.
The pertaining sentence was specified
If all the corrections are incorporated, the statistic is revised and the presentation is improved the paper could be accepted for its publication in Vaccines.
Best Regards,
The reviewer.
Additional clarifications were done all over the manuscript and highlighted.
We look forward to hearing from you in due time regarding our submission and to respond to any further questions and comments you may have.
Sincerely,
Paul Thöne

Reviewer 3 Report
Comments and Suggestions for Authors
Thone and colleagues carried out a serosurveillance study on cancer patients, who have either received radiotherapy (RT) or RT combined with chemotherapy. This study is important and meaningful, but some major concerns must be addressed, and the results must be clearly presented.
Major concerns
1. Binding antibodies do not protect people from SARS-CoV-2 infection, while neutralizing antibodies do.
2. Precise wordings and careful proof-reading are required. The meaning of some sentences is not clear.
a. Line 26, 28, 75, 76, 92-93, 120-122, 142-144, 199-201, 291, 300
3. Line 124: data and method of IFN-gamma release assay are presented.
4. Section 3.2: it is highly recommended to present the result in a figure.
5. Table 2: what does the number in the rows with “n day 0, 7,…” mean? Why was the number decreased with time?
6. Section 3.7 and Table 1: Side effect of patients in cohort 1 and 2 was assessed, while cohort 3 was not. How could the frequencies of side effects in cohort 3 be measured?
Comments on the Quality of English LanguageI don't have further comments.
Author Response
Dear Reviewer,
we hope this letter finds you in the best spirits!
Thank you for gciving us the opportunity to submit a revised draft of our manuscript Immunogenicity parameters of cancer patients receiving the mRNA vaccine BNT162b2 while obtaining radiotherapy: a longitudinal cohort evaluation. We appreciate the time and effort for your reviewing and the dedication of your accurate and valuable feedback to enhance the quality of our work. We have been able to incorporate changes to reflect most of the provided suggestions. The changes within the manuscript are highlighted.
Here is a point-by-point response to your comments and concerns:
Thone and colleagues carried out a serosurveillance study on cancer patients, who have either received radiotherapy (RT) or RT combined with chemotherapy. This study is important and meaningful, but some major concerns must be addressed, and the results must be clearly presented.
Major concerns
- Binding antibodies do not protect people from SARS-CoV-2 infection, while neutralizing antibodies do.
Notably, an assessment of neutralizing antibodies, as presented in the approval study of BNT162b2 is gold standard for assessing the vaccination efficacy. The choice to consider binding antibodies is based on the limited laboratory resources in our clinical setting. Our investigation emerged out of a retrospective assessment, which enabled affected patients to monitor their immune response in order to immunization with available methodes. In our case, this was the assessment of binding antibodies which is, as mentioned in the introduction, a reliable parameter of the humoral immune response
- Precise wordings and careful proof-reading are required. The meaning of some sentences is not clear.
- Line 26, 28, 75, 76, 92-93, 120-122, 142-144, 199-201, 291, 300
We kindly thank you for the advice, the pertaining passages were revised.
- Line 124: data and method of IFN-gamma release assay are presented.
Line 124 provides information about the T-cellular essay, which was performed in an external laboratory department using an on-site essay.
- Section 3.2: it is highly recommended to present the result in a figure.
Thank you for the recommendation. The data is now portrayed more comprehensive in a figure.
- Table 2: what does the number in the rows with “n day 0, 7,…” mean? Why was the number decreased with time?
“n” specifies the number of tested subjects in the pertaining measurement day. This information is now additionally declared the figure key below.
- Section 3.7 and Table 1: Side effect of patients in cohort 1 and 2 was assessed, while cohort 3 was not. How could the frequencies of side effects in cohort 3 be measured?
Information referring to side effects in cohort 3 are already denoted in the text passage of the results section, as well as in the former table 1 (now appendix). If data is unclear in another way, do not hesitate to highlight the affected passage again.
Additional clarifications were done all over the manuscript and highlighted.
Clarifications were done and highlighted.
We look forward to hearing from you in due time regarding our submission and to respond to any further questions and comments you may have.
Sincerely,
Paul Thöne

Round 2
Reviewer 1 Report
Comments and Suggestions for Authors
The authors have addressed all the reviewer's concerns about the tiny samples, reduced the amount of inappropriate analysis and re-edited the entire text.
Author Response
Dear Reviewer,
we hope this letter finds you in the best spirits!
Thank you for your time and effort in providing us a feedback. Your suggestions helped a lot to enhance the quality of our manuscript.
With regards,
For all authors,
Paul Thöne

Reviewer 2 Report
Comments and Suggestions for Authors
Dear Authors,
I insist that you could do better job in the statistic section. Anyhow the manuscript is improved.
All the best,
The reviewer.
Author Response
Dear Reviewer,
we hope this letter finds you in the best spirits!
After careful consideration, we desist from making use of different statistical methods. As mentioned, we reckon our methodology as conclusive and best suited in this particular situation.
Nevertheless, we want to thank you for your time and effort in providing us a feedback. Your suggestions helped a lot to enhance the quality of our manuscript.
With regards,
For all authors,
Paul Thöne

Reviewer 3 Report
Comments and Suggestions for Authors
The manuscript is improved and most of the concerns have been addressed. However, as mentioned in the previous review report, binding antibody is conceptually different from neutralizing antibody. The limitation and the reason why binding antibody presented in this manuscript should be presented.
Comments on the Quality of English LanguageI don't have any further comments.
Author Response
Dear Reviewer,
we hope this letter finds you in the best spirits!
As mentioned before, we share your opinion, that an analysis of neutralizing antibodies is the best suited to assess the efficacy of vaccines. In our investigation, an analysis of neutralizing antibodies was not provided as a subject of research. We evaluated binding antibodies, which nevertheless show advantages, such as the easier clinical availability or the international standardization and comperability (BAU).
Thank you for your time and effort in providing us a feedback. Your suggestions helped a lot to enhance the quality of our manuscript.
With regards,
For all authors,
Paul Thöne
